# Effects of Electron Beam Irradiation on Mechanical and Tribological Properties of PEEK

**DOI:** 10.3390/polym15061340

**Published:** 2023-03-07

**Authors:** Bayan Kurbanova, Kazybek Aimaganbetov, Kanat Ospanov, Kairat Abdrakhmanov, Nurkhat Zhakiyev, Bauyrzhan Rakhadilov, Zhuldyz Sagdoldina, Nurlan Almas

**Affiliations:** 1Department of Physics, School of Sciences and Humanities, Nazarbayev University, Astana 010000, Kazakhstan; 2Institute of Hydrogen Energy, International Science Complex Astana, Astana 010000, Kazakhstan; 3Department of Science and Innovation, Astana IT University, Astana 01000, Kazakhstan; 4Department of Physics, M. Utemisov West Kazakhstan University, Uralsk 09000, Kazakhstan; 5Research Center Surface Engineering and Tribology, Sarsen Amanzholov East Kazakhstan University, Oskemen 070000, Kazakhstan

**Keywords:** polyetheretherketone, tribological test, friction coefficient, wear rate, microhardness

## Abstract

In this work, the mechanical and tribological characteristics of polyetheretherketone (PEEK) sheets were enhanced by electron beam irradiation. PEEK sheets irradiated at a speed of 0.8 m/min with a total dose of 200 kGy achieved the lowest specific wear rate of 4.57 ± 0.69 (10^−6^ mm^3^/N^−1^m^−1^), compared to unirradiated PEEK with a rate of 13.1 ± 0.42 (10^−6^ mm^3^/N^−1^m^−1^). Exposure to an electron beam at 9 m/min for 30 runs, with a dose of 10 kGy per run for a total dose of 300 kGy, resulted in the highest improvement in microhardness, reaching 0.222 GPa. This may be due to the decrease in crystallite size, as indicated by the broadening of the diffraction peaks in the irradiated samples. According to the results of thermogravimetric analysis, the degradation temperature of the irradiated samples remained unchanged at 553 ± 0.5 °C, except a sample irradiated at dose 400 kGy, where the degradation temperature shifted towards a lower position of 544 ± 0.5 °C. Differential scanning calorimetry results revealed that the melting temperature ($T_{m}$) of the unirradiated PEEK was about 338 ± 0.5 °C, while a high temperature shift of the $T_{m}$ was observed for the irradiated samples.

## 1. Introduction

Polyetheretherketone (PEEK) is a semi-crystalline thermoplastic material with excellent resistance to wear, fatigue, creep, and chemicals, as well as a high melting point, mechanical toughness, and strength. This polymer has a variety of uses in industries such as automotive, aerospace, oil and gas, medical, and others, due to its remarkable physical and chemical properties and low cost [1,2,3]. Numerous studies have shown that functionalized PEEK is suitable for use in PEFC membranes [4,5,6]. It appears that the PEEK polymer’s improved properties will allow for the discovery of new uses for the material as well as improvements in its current ones. Changing the length of the polymer chains, making branched chains from linear polymer chains, crosslinking the polymer chains, and adding plasticizers to the polymer are the basic methods for modifying and fine-tuning a polymer’s properties [7,8,9,10,11].

Numerous researches have examined how PEEK and its composites react to various radiation sources, such as heavy ions [12,13,14], gamma rays [15,16], UV radiation [17], or combined with aging [18] and electron irradiation [19,20] since the polymer was first commercialized in 1978. These methods are very effective for controlling various properties of materials including polymers. Radiation sources are an effective way to process materials without the need for industrial chemicals. Irradiation can cause microstructural crosslinking and/or polymer chain scission, altering the mechanical properties of polymers.

The effect of electron beam irradiation on the mechanical properties of a carbon fiber-reinforced PEEK nanocomposite at low and high temperatures was investigated [21]. The viscoelastic measurement made it clear that the high-temperature shift in the glass transition of the PEEK matrix generated by radiation-induced crosslinking was what caused the improvement in mechanical properties at high temperatures. The effect of electron beam irradiation on the mechanical relaxation of semi-crystalline PEEK was studied by measuring the dynamic viscoelastic properties over a wide temperature range [22]. The authors of this study found that both crosslinking and scission of molecular chains occur during irradiation. In Ref. [23], it was demonstrated that 1.0 MeV electron irradiation improved the phase morphology, interfacial adhesion, and mechanical properties of a PTFE/PEEK blend. In [24], it was discovered that electron irradiation with a dose of more than 20 MGy reduced the thermal and mechanical properties of PEEK sheets. According to the authors of this study, crosslinking acts as PEEK’s primary mechanism during electron beam irradiation. The mechanical tests used in this research demonstrated that PEEK films have high radiation resistance. Tensile strength, Young’s modulus, and peak strain all rose with radiation dosage, whereas permanent elongation decreased.

High doses of high-energy electron irradiation, however, have the potential to negatively impact the characteristics of polymers by altering their structure. In an electron beam exposed to radiation with a high voltage of 1.5 MeV and a high absorbed dosage of 1000 Mrad, radiation damage to the PEEK matrix was observed [25]. The effect of irradiation with 170 keV protons on the melting and crystallization temperatures of PEEK was studied by differential scanning calorimetry [26]. The results of DSC showed that the crystallinity of the polymer after irradiation decreases from 17% to 13%. The crystal structure of PEEK changed and its crystallinity decreased after irradiation, leading to a reduction in both the enthalpy of crystallization and the enthalpy of melting. The tribological properties of PEEK irradiated with gamma rays up to 20 MGy were studied in [27]. X-ray photoelectron spectroscopic analysis showed an increase in the percentage of C=O and O–C=O functional groups on the surface of irradiated PEEK samples. Irradiation-induced oxidation was the main form of PEEK surface degradation. Friction experiments have shown that the static friction coefficient of the irradiated PEEK surface increases due to the presence of a surface oxide layer, while the stationary dynamic friction coefficient of the irradiated PEEK surface gradually decreases at a dose of more than 1 MGy. Surface oxidation products played a role in three-body rolling, which led to a drop in the dynamic friction coefficient. However, the corresponding wear rate increased by an order of magnitude compared to the non-irradiated sample. According to DSC tests, UV irradiation does not affect the polymer crystallinity and melting point of the tested PEEK fibers, but significantly increases the crosslinking rate, making the fibers brittle [28]. Sheet samples of PEEK with a thickness of 0.22 mm were irradiated with an electron beam with an energy of 10 MeV to doses of 1500 kGy in air at room temperature to study the process of radical formation [29]. Using electron paramagnetic resonance spectroscopy at ambient temperature and in liquid nitrogen, the radical anion, phenoxy radical, and phenyl peroxy radical were identified. The authors of this work concluded that despite the formation of radicals due to core cleavage and degradation, the macroscopic characteristics remained unchanged upon irradiation up to a dose of 1500 kGy. In particular, for the studied polymer, a slight decrease in elasticity was observed at doses less than 600 kGy, accompanied by fluctuations in tensile strength ranging from 101 to 107 MPa and an average elongation at break of about 260%, varying within a standard deviation of ±10%. The decrease in Young’s modulus was due to gradual progression of chain scissioning with increasing dose, and the increase in Young’s modulus for doses above 600 kGy was due to the growth of the crystalline phase, which was confirmed by DSC measurements.

The purpose of this work was to investigate the effect of relatively low doses of high-energy electron irradiation on the mechanical and tribological characteristics of PEEK using a pulsed linear electron accelerator of the ILU-10 type. The tribology tests and microhardness measurements were used to analyze the specific wear rate and Marten’s hardness of PEEK. X-ray scattering, Differential Scanning Calorimetry (DSC), and Thermogravimetric Analysis (TGA) were applied to analyze the structure, melting and glass transition temperatures, as well as crystallinity change of the PEEK after electron beam irradiation.

ILU-10 accelerators have been in use for many years in industrial and research radiation technology facilities throughout several nations [30]. The circuit configurations and long-term continuous operation of the accelerators under industrial production circumstances are made possible by the design of the accelerators. In linear particle accelerators, electrons are accelerated in vacuum along the axis of the accelerator by a longitudinally directed electric field. ILU-type accelerators are referred to as high-frequency accelerators because they use an alternating high-frequency electric field to accelerate electrons. The operating frequency of the ILU-10 accelerator is 118 MHz. The ILU-10 accelerator uses a triode electron gun located directly in front of the accelerating gap. The use of a control voltage on the electron gun makes it possible to quickly control the beam current and reduce the injection phase angle, which significantly reduces the spread of the electron energy in the beam. ILU-type accelerators have advantages and disadvantages in relation to the most common type of accelerators, linear accelerators. The advantages of the ILU-10 accelerator include low cost, simplicity, and reliability of the accelerating structure, high energy of accelerated electrons combined with a sufficiently high beam power (tens of kilowatts). The accelerator quickly reaches its operating parameters after the start, and the location of the controlled triode electron gun directly in front of the accelerating gap of the resonator makes it possible to quickly adjust the pulsed beam current and thus its average power in a fairly wide range. The presence of some energy spread of electrons in the beam smooths out the uneven distribution of the dose inside the processed products. The disadvantages of ILU accelerators are the relatively low conversion factor of the electric power consumed from the network into the beam power, as well as the limited maximum power of the accelerators (up to 50 kW).

## 2. Materials and Methods

### 2.1. Electron Irradiation

We investigated PEEK plates made by Ensinger (Ensinger Poly Tech Inc., Huntersville, NC, USA), (based on Victrex^®^ PEEK 450 G or Solvay’s KetaSpire^®^ KT-820 polymer) with a 10 mm^2^ surface area and a 5 mm thickness. At the Park of Nuclear Technologies in Kurchatov, Kazakhstan, the ILU-10 industrial electron pulse accelerator equipment was used to irradiate the polymer plates with fast electrons at room temperature. The electron beam current, and energy of the accelerator were tuned to 6.87 mA and 2.7 MeV, respectively. The sample surface electron beam diameter and pulse length were both tuned to 10 mm and 0.4–0.5 ms, respectively. A total of 400 kGy was applied to the polymer plates.

### 2.2. Tribological Test

The usual ball-and-disk technique was used to conduct the friction-sliding tribological test on a TRB3 tribometer (Anton Paar, Oslo, Norway) under strict dry friction specifications and constant room temperature. As a countersample, a steel ball with a 100Cr6 coating and a 6.0 mm diameter was used. The tests were run at a linear speed of 5 cm/s on a 2 mm radius under a load of 10 N. Over a test distance of 50 m, the change in the coefficient of friction was measured.

### 2.3. Wear Rate

The degree of polymer wear was calculated during tribological testing using the Profiler 130. The profilometer operates by scanning the surface irregularities with a probe while an inductive sensor moves along the surface, transforming the ensuing mechanical vibrations of the probe. Four cross-sectional profiles were measured at equal intervals for each wear mark and used to determine wear volume loss estimates in line with ASTM G99-05 specifications. The measurement limit Rt was 50 m, and the tracing speed was 0.25 mm/s (fine scale).

### 2.4. Microhardness

The Fisherscope HM2000 S measuring system was used to calculate the microhardness in compliance with DIN EN ISO 14577-1 specifications. The displacements were measured with an accuracy of 0.1 nm, and the load setting accuracy was 4 mg. The microhardness measurement error was less than 2% of the observed value. The indenter approached at a 2 m/s speed. The testing load range was between 1 and 2000 mN. The WIN-HCU instrument software (WIN-HCU® Version 7.1 are registered trademarks of Helmut Fischer GmbH Institut für Elektronik und Messtechnik, Sindelfingen, Germany) was used for the test findings’ first processing. As an indenter, a tetrahedral Vickers diamond pyramid with a 136° angle was employed.

### 2.5. X-ray Diffraction (XRD)

To assess the effect of electron beam irradiation on the structural-phase state of PEEK, X-ray phase analysis was performed using an X’Pert PRO diffractometer (PANalytical, Almelo, The Netherlands). Shooting modes: diffraction angle from 10° to 45°; scan step 0.03; exposure time 0.5 s; radiation: CuKα; voltage and current: 45 kV and 30 mA.

### 2.6. DSC and TGA Analysis

The melting and glass transition temperatures were measured by Differential Scanning Calorimetry (DSC) and Thermogravimetric Analysis (TGA) by a Simultaneous Thermal Analyzer (STA) 6000 (PerkinElmer, Waltham, MA, USA) in a nitrogen atmosphere, while the samples’ test temperature ranged from 30 to 800 °C at a heating rate of 10 °C/min.

## 3. Results and Discussion

### 3.1. Electron Irradiation

Irradiation modes were linked to variations in dose, which depended on the number of runs, i.e., the movement of the samples relative to the electron beam on the moving table. This took into account the cumulative effect of electron beam irradiation on polymer properties. Table 1 lists the electron beam irradiation modes used on the PEEK samples.

In our previous study, monte CArlo SImulation of electroN trajectory in sOlids (CASINO) software was used to estimate the penetration depth of electrons, and backscattered and transmitted energies in PEEK samples [31]. The simulation of PEEK samples exposed to 2.7 MeV electrons included the particle collision process, the maximum depth to which the electron trajectories might penetrate the sample, the energy of the transmitted electrons, and the energy of the backscattered electrons. It was found that the electrons with an energy of 2.7 MeV completely passed through the sample. Most of the electrons were transmitted rather than absorbed in the irradiated sample, and their energies were ~1.5 MeV. During incident, the backscattered electrons*’* highest energy was 2.1 MeV.

### 3.2. Effects of Electron Beam Irradiation on Friction Coefficient, Wear Rate, and Microhardness

Figure 1 shows the variation of the coefficient of friction of PEEK with changes in radiation doses under constant load. It is evident that, except for PEEK-1, the friction coefficients of all samples decreased after exposure to irradiation. At the very beginning of the test, the coefficient of friction was minimal and was about 0.10 for non-irradiated PEEK and PEEK-1; 0.075 for PEEK-5; 0.05 for PEEK-3; and about 0.03 for PEEK-4, PEEK-6, and PEEK-7. After about 10-15 m sliding distance in the case of non-irradiated PEEK, PEEK-1, PEEK-3, PEEK-4, PEEK-5, and PEEK-6, μ rapidly increased to 0.28; 0.30; 0.23; 0.21; 0.27; and 0.26, respectively, and this value did not change until the end of the test. In the case of PEEK-7, µ slowly increased to 0.22 until the end of the test. In the case of PEEK-2, μ rapidly increased to 0.18 at a sliding distance of about 30 m and this value of the friction coefficient did not change until the end of the test. Thus, PEEK-2 irradiated with a dose of 100 kGy demonstrated the lowest coefficient of friction of all the studied samples. The friction coefficient value for unirradiated PEEK obtained in our work is consistent with the literature data. The friction coefficient values of PEEK highlighted in the literature are shown in Table 2.

In our previous work, the roughness analysis of pristine and irradiated PEEK was performed using the AFM technique [31]. Irregular grooves and bumps resulting from sample polishing were found on the surface of a non-irradiated sample. The average surface roughness (Ra) of the pristine PEEK was about 265 nm. Irradiation with PEEK at a dose of 200 kGy led to a significant increase in the number of irregularities, while their length decreased, and their depth, on the contrary, increased. The average surface roughness (Ra) of the irradiated PEEK increased significantly and amounted to 750 nm. With an increase in the irradiation dose to 400 kGy, the surface of the sample was smoothed, and the average surface roughness decreased to 665 nm. Such changes in the surface roughness of the samples were apparently associated with irradiation damage, as well as with the processes of evaporation and partial melting of their near-surface region.

Under wear conditions, PEEK polymers can form a protective transfer film. The exact mechanism of the PEEK transfer film formation is unknown. Transfer film formation is considered an integral part of the wear resistance of PEEK. Surface roughness has been found to be a determining factor in PEEK wear and its relationship to transfer film formation has been extensively studied [40,41]. A study of the effect of contact surface roughness on PEEK revealed that minimal wear occurs at a roughness of 0.15 µm [42]. It has been shown in [43] that transfer films tend to deposit on one side of the surface asperities of the specimen in overlap sliding. All plates were ground to a surface roughness (Ra) of 0.5 µm, the height of the asperities and the distance between them corresponded to the Gaussian distribution. The authors of the work suggested that there is some asperity critical height and distance at which the deposited material cannot overlap. In the case of irregularities of sufficient size, the removed material accumulated and agglomerated into large islands. Wear residue collected on the surface of higher asperities, but no continuous transfer film was formed. Likewise, large gaps between the asperities prevented material from depositing on every bump.

In our case, a quite uniform distribution of surface roughness was observed for samples irradiated to doses of 200 kGy, as shown in Figure 2. We believe that the specifics of transfer film formation were the key factors that influenced the results of our tribological tests. Wear leads to deformation of the polymer, accompanied by slip, stretch, and orientation of molecular chains in the direction of the applied stress [44]. Low wear may be attained by stopping this chemical reconfiguration using irradiation crosslinking [45].

The wear intensity was used as a metric to compare the tribological properties of the samples before and after electron beam irradiation. The wear rate under the impact of the tip was estimated based on the volume of material displaced during the test, using the following Formula (1):(1)$$I=\frac{V}{F\times l}$$

Here, *I*, *V*, *F*, and *l* were the specific wear rate, nominal pressure, worn part volume, and friction path, respectively. By measuring the wear track profile, we were able to estimate the degree of polymer wear that resulted from the tribological test. Figure 3 displays the results of our estimation of the wear rate of PEEK before and after electron beam irradiation.

In comparison to the initial sample, the study’s findings indicated that PEEK-1 had a high wear intensity (irradiation dose of 50 kGy). PEEK-4 showed the highest wear resistance of 4.5 × 10^−6^ mm^3^N^−1^m^−1^; the wear resistance of samples PEEK-3, PEEK-2, and PEEK-5 were also markedly improved and amounted to 7.25 × 10^−6^ mm^3^N^−1^m^−1^, 7.5 × 10^−6^ mm^3^N^−1^m^−1^, and 8 × 10^−6^ mm^3^N^−1^m^−1^, respectively.

To characterize the microhardness of the sample, we chose the hardness HM on the Martens scale as the parameter. The hardness value for a variety of materials can be calculated by taking into account both plastic and elastic deformations during the measurement of HM. The formula for calculating Martens hardness, as stated in (2), involves dividing the test load’s current value by the indenter’s cross-sectional area at a specified distance from the top:(2)$$HM=\frac{F}{A_{s}\left(h\right)}=\frac{F}{26.43h^{2}\;}$$

Here, *HM*, *F*, *A_s_(h)*, and *h* are the Marten’s hardness, test load, indenter cross-sectional area, and indenter penetration depth, respectively.

The measurements were done using the conventional loading and unloading techniques (increase and decrease in force). At a maximum force of 500 mN, the indenter penetrated the sample for a certain amount of time (load increase), after which it was removed for a predetermined amount of time (force decline), which was equivalent to 20 s.

PEEK specimen loading patterns are shown in Figure 4. After five repeated measurements, the loading–unloading diagrams of the samples matched up without any notable shifts. This would suggest that irradiation has a uniform influence on polymer structure.

The measurements of the Martens hardness of PEEK samples before and after electron beam irradiation are shown in Figure 5. The microhardness of all samples increased as a result of irradiation. The PEEK-5 sample that received a dosage of 300 kGy of irradiation was found to have the highest microhardness measurement of 0.222 GPa, while the microhardness of the unirradiated PEEK was minimal and amounted to 0.191 GPa.

### 3.3. XRD Spectra

Typically, the properties of polymers change when exposed to high intensity radiation. This occurs due to a complicated and random process when the polymer interacts with energetic electrons. The energies in action are significantly higher than the energy required for every electron to bond with an atom. This results in the breaking of bonds and the formation of radicals due to the transfer of energy from the electrons to the molecule. The combination of radicals can lead to the formation of crosslinks, end-links, or disproportionate reactions, as well as gaseous molecules. PEEK diffraction patterns before and after electron beam irradiation are shown in Figure 6. The maximum angles of the four diffraction peaks are about 18.7°, 20.7°, 22.6°, and 28.7°. The nearly identical diffraction patterns across a range of irradiation doses show that the PEEK crystal structure was only minimally affected by electron beam irradiation. At various irradiation dosages, there were modest variations in the height and width of the diffraction peaks, which could indicate a change in the degree of crystallinity (the proportion of amorphous to crystalline phases) of PEEK. We evaluated the basic diffraction spectral parameters of samples of the pristine, unirradiated PEEK, PEEK-3, and PEEK-5, whose microhardness increased the most when compared to the other irradiated samples. Table 3, Table 4 and Table 5 contain the details of the samples*’* diffraction spectra. The height of every peak dropped as a result of EBI, as shown in Table 4 and Table 5. From Table 4 and Table 5 one can also observe the broadening of the diffraction lines and the full widths at half maximums (FWHM) following EBI. EBI had almost no impact on the interplanar spacing parameter d.

In more detail, let us examine the broadening of diffraction peaks. The integral width analysis of diffraction peaks is a commonly used method for determining nanostructure parameters due to its simplicity, especially when using FWHM instead of integral width. In 1918, Scherrer demonstrated that small crystallites result in the broadening of diffraction lines, and that the width of the diffraction line profile is inversely proportional to the size of the crystallites in a sample:(3)$$\beta_{s}=\frac{\lambda }{D\;Cos\theta }$$

Here, *D* is the effective crystallite size and *β_S_* is the integral peak width.

The Hall–Petch relation states that a decrease in grain size leads to an increase in the yield strength and microhardness of a polycrystalline material.
(4)$$\sigma_{y}=\sigma_{0}+\sigma \;\frac{k_{y}}{\sqrt{d}}$$

Here, *σ_y_* is the yield stress, *σ*_0_ is a materials constant for the starting stress for dislocation movement (or the resistance of the lattice to dislocation motion), *k_y_* is the strengthening coefficient (a constant specific to each material), and *d* is the average grain diameter. Theoretically, a material can be made infinitely strong by making its grains infinitely small.

We believe that the increase in the microhardness of the irradiated samples that we observed was associated with a decrease in the size of crystalline regions or crystallites. The crystal structure of polyaryl ether ketones (PAEK), studied using high-resolution electron microscopy, showed that PAEK is built from granular crystalline blocks with a size of 75–145 nm that self-organize along the radial direction [46]. The study also showed that granular crystalline blocks were composed of small primary crystals approximately 20–30 nm in size, united by a secondary structure. The secondary structure turned out to be less ordered and was predominantly removed during chemical etching, while the primary crystals remained intact. PEEK belongs to the PAEK family and is a class of semi-crystalline polymers that exhibits both crystalline and amorphous phases in its structure. Semicrystalline PEEK has crystalline lamellae as its structure, which is a good first approximation [47]. In general, defects are the major outcome of electron beam irradiation on a crystalline region. The original crystalline structure typically gradually disintegrates with high radiation doses. However, after exposure to radiation, some polymers show an initial increase in crystallinity [48,49].

### 3.4. TGA and DSC Analysis

The TGA and DSC curves of the pristine and irradiated PEEK samples at the heating stage are shown in Figure 7 and Figure 8, respectively. The TGA plots show that there was no significant change in the degradation temperature of the PEEK before and after electron irradiation. The onset degradation temperature was 553 °C. However, it is evident that there was a significant difference in the degradation temperature of PEEK-7 from the other samples, where the temperature decreased to 544 °C. This can be attributed to random chain scissions between aromatic rings and ether or ketone bonds independently of the nitrogen atmosphere [50,51].

DSC analysis was performed to understand the glass transition temperature ($T_{g}$) and melting temperature ($T_{m}$) changes of the PEEK samples that were subjected to electron beam irradiation. In Figure 8b we can observe that there was no change in glass transition temperature for the samples before and after irradiation. The glass transition temperature was 150 °C. However, the changes were observed in melting temperature for the irradiated samples in comparison with the pristine PEEK. From Figure 7 it can be clearly seen that the melting peak of the pristine PEEK was about 338 °C, while the melting temperature of the irradiated samples except PEEK-7 shifted to a higher temperature range. The thermal stability of the irradiated samples increased during crosslinking due to the stiffening of the polymers [26]. Figure 8d shows the maximum weight loss temperature of PEEK samples during the heat flow process.

## 4. Conclusions

In this work, through tribological tests, microhardness measurements, as well as X-ray diffraction, DSC, and TGA analyses, the effect of irradiation with a high-energy electron beam with an energy of 2.7 MeV and doses from 50 to 400 kGy on the mechanical and tribological properties of PEEK sheets was studied. It was established that irradiation conditions can significantly affect the properties of polymers. In particular, in our experiments, the total dose received by the samples during irradiation was varied by changing the speed of movement of the samples under the electron beam, the number of runs, and the irradiation dose per run. Thus, a significant improvement in the wear resistance of the polymer irradiated with doses of 200 kGy was observed. In addition, an increase in Martens microhardness was observed for all irradiated samples. The decrease in the size of the crystalline regions of the PEEK sheets, which was established from the broadening of the diffraction lines in the spectra of irradiated samples, may be one of the reasons for the increase in the microhardness of the samples. The cumulative effect of the competing processes of splitting and crosslinking of PEEK macromolecules under the action of a high-dose electron beam seems to be another factor that induced the improvement in mechanical and tribological properties. The results of the thermogravimetric analysis (TGA) showed that there was no significant change in the decomposition temperature of the samples, except for the sample that received the maximum radiation dose of 400 kGy, in which the decomposition temperature decreased slightly. This may have been due to random chain breaks between aromatic rings and ether or ketone bonds. The shift of the melting temperature towards higher temperatures, except for the sample that received the maximum irradiation dose of 400 kGy, was shown by the DSC results. Apparently, the thermal stability of the irradiated samples increases upon crosslinking due to the solidification of the polymer.

## Figures and Tables

**Figure 1 polymers-15-01340-f001:**
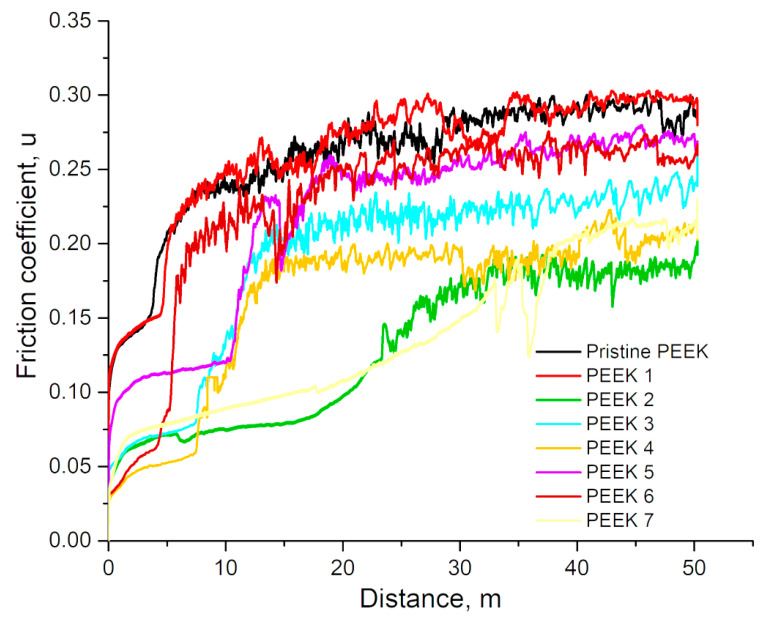
The coefficient of friction under constant load as functions of radiation dose of PEEK.

**Figure 2 polymers-15-01340-f002:**
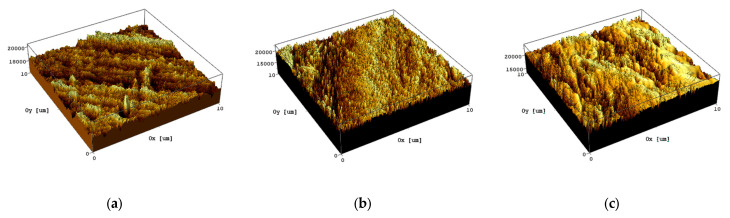
3D surface morphology of pristine (**a**) and irradiated PEEK: (**b**) 200 kGy; (**c**) 400 kGy.

**Figure 3 polymers-15-01340-f003:**
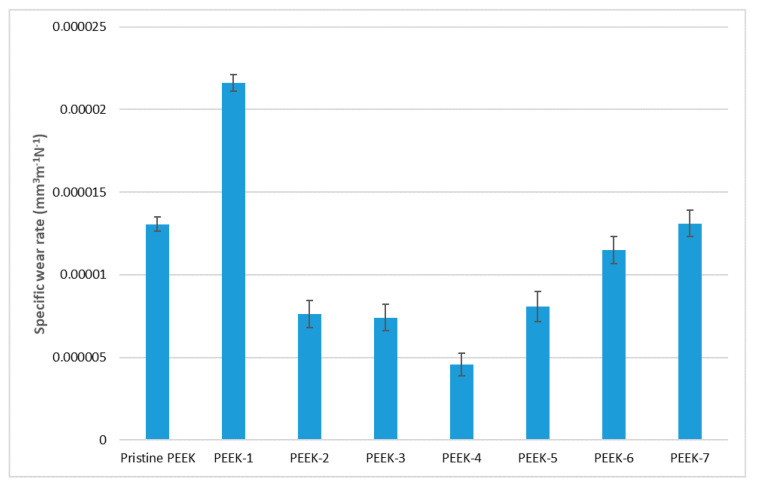
Variations of PEEK specific wear rate.

**Figure 4 polymers-15-01340-f004:**
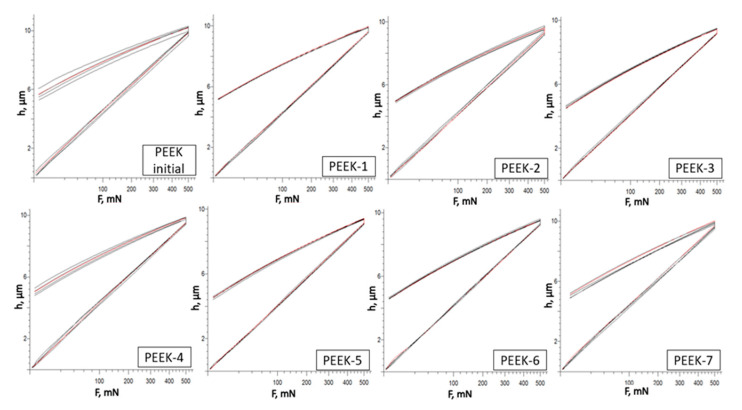
Load–displacement data for PEEK samples.

**Figure 5 polymers-15-01340-f005:**
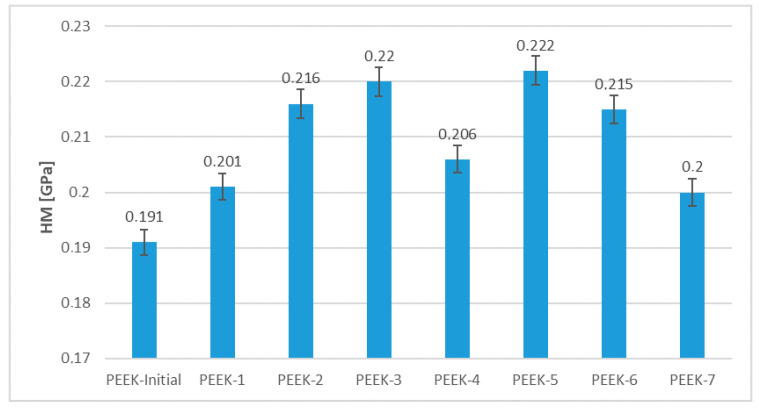
Variations of PEEK microhardness.

**Figure 6 polymers-15-01340-f006:**
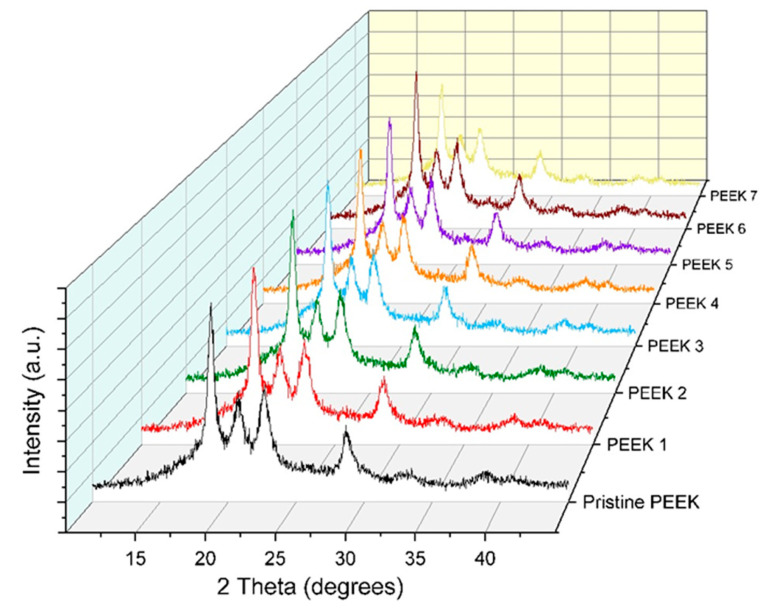
X-ray diffraction patterns of PEEK before and after electron beam irradiation.

**Figure 7 polymers-15-01340-f007:**
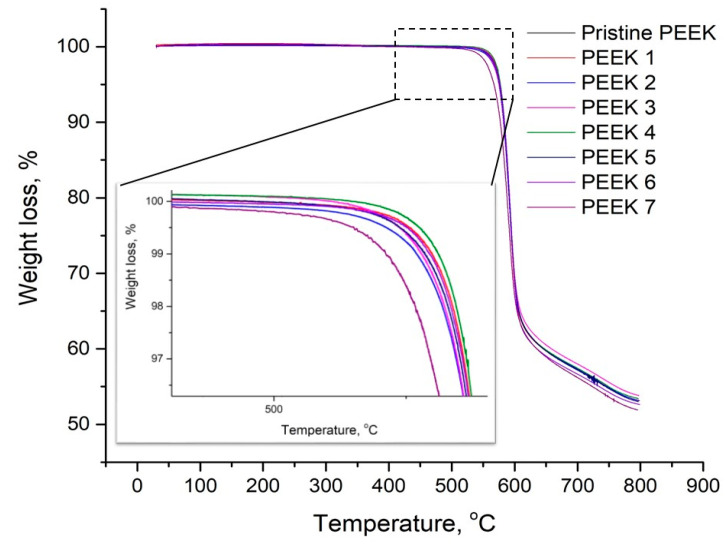
TGA curves of the pristine and irradiated PEEK samples.

**Figure 8 polymers-15-01340-f008:**
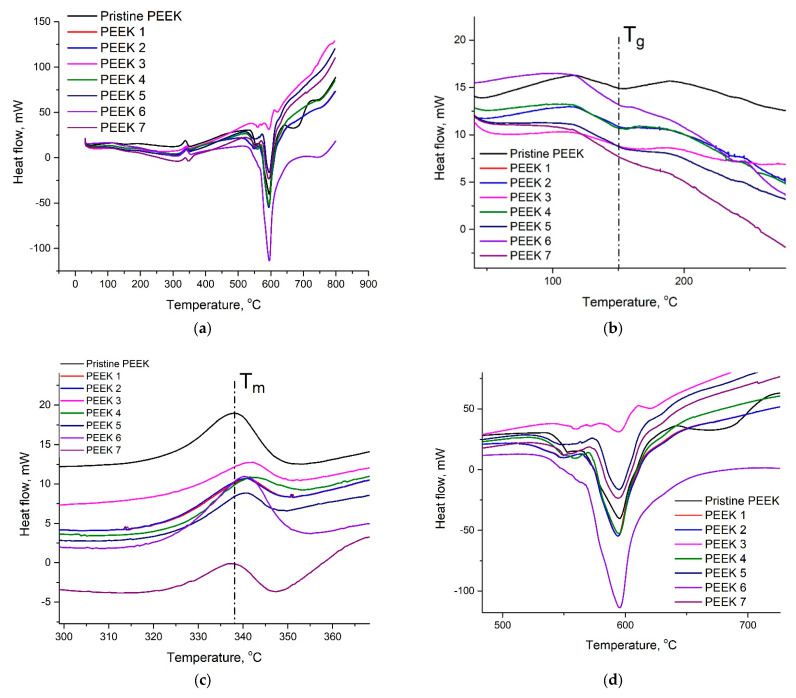
DSC curves of the pristine and irradiated PEEK samples (**a**). (**b**) Glass transition, (**c**) melting, and (**d**) weight loss temperatures.

**Table 1 polymers-15-01340-t001:** Electron beam irradiation modes of PEEK samples at the ILU-10 accelerator.

Number of Samples	Energy, MeV	Beam Current, mА	Velocity, m/min	Number of Runs	Dose per Run, kGy	Total Dose, kGy
PEEK-1	2.7	6.84	9	5	10	50
PEEK-2	2.7	6.84	9	10	10	100
PEEK-3	2.7	6.84	9	20	10	200
PEEK-4	2.7	6.84	0.8	2	100	200
PEEK-5	2.7	6.84	9	30	10	300
PEEK-6	2.7	6.84	0.8	3	100	300
PEEK-7	2.7	6.84	9	40	10	400

**Table 2 polymers-15-01340-t002:** Friction coefficient value of PEEK.

µ	Wear Rate (10^−5^ mm^3^/Nm)	Reference
0.3	0.38	[32]
0.33	1.2	[33]
0.15–0.20	0.004	[34]
0.42		[35]
0.3–0.42	0.1–04	[36]
0.41	1.2	[37]
0.34–0.42	1.1–2.5	[38]
0.49–0.61	0.16–35	[39]
0.18–0.30	0.45–2.2	our results

**Table 3 polymers-15-01340-t003:** Basic diffraction spectral parameters of pristine PEEK.

Peak Position	Height	FWHM	d [Å]	Intensity [%]
18.7316	504.15	0.3542	4.73732	100.00
20.7309	191.75	0.4133	4.28474	38.03
22.6433	240.63	0.4133	3.92701	47.73
28.6634	133.99	0.2952	3.11445	26.58

**Table 4 polymers-15-01340-t004:** Basic diffraction spectral parameters of PEEK-3.

Peak Position	Height	FWHM	d [Å]	Intensity [%]
18.7233	487.15	0.3542	4.73941	100.00
20.7102	188.45	0.4133	4.28898	38.68
22.6043	223.03	0.7085	3.93369	45.78
28.7211	120.03	0.5904	3.10833	24.64

**Table 5 polymers-15-01340-t005:** Basic diffraction spectral parameters of PEEK-5.

Peak Position	Height	FWHM	d [Å]	Intensity [%]
18.7153	473.77	0.4133	4.74140	100.00
20.7096	185.59	0.4723	4.28911	39.17
22.6457	222.31	0.3542	3.92660	46.92
28.7552	124.14	0.2952	3.10472	26.20

## Data Availability

Not applicable.

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
