# Peer review of "Effects of Electron Beam Irradiation on Mechanical and Tribological Properties of PEEK"

_polymers, 2023, doi:10.3390/polym15061340_

Round 1

Reviewer 1 Report

 This paper is a continuation of the previous work by these authors (Ref. 17). Coefficient of friction, wear resistance and microhardness were investigated after electron beam irradiation. X-ray diffraction was also studied, however no definite conclusions are made. After reading the manuscript, a sense of incompleteness remains. Each measured parameter has either a maximum or minimum with increasing radiation dose, while no definite explanation is given. On p. 8 two mutually exclusive hypotheses are presented - "crystalline structure gradually disintegrates with high radiation doses" and "some polymers show an increase in crystallinity". In Ref. 17 the authors explained the results by scission of macromolecular chains, while in present paper the explanation is crosslinking of polymer chains. These issues should be attended to in revised version.

Author Response

Dear reviewer, we have taken into account all your comments and made the necessary changes and additions to the article. We reanalyzed the results of X-ray diffraction of the samples. The broadening of diffraction lines observed in the spectra of irradiated samples allowed us to make an assumption about a decrease in the size of their crystalline regions. As is known, the reduction of crystallites of polycrystalline materials increases their microhardness. These data are described in more detail in the supplemented and corrected article. However, we cannot give an exact quantitative relationship between the processes of breaking and crosslinking of molecular chains of samples under the influence of electron irradiation at the moment. We only assume that the cumulative effect of these two competing mechanisms leads in some cases to an improvement in the properties of the polymer, and in some cases to degradation of its properties. We hope to conduct additional studies of the fine structure of samples by EPR, NMR and other methods and write another article in which we can quantify these processes.

Reviewer 2 Report

The manuscript "Effects of electron beam irradiation on mechanical and tribological properties of PEEK" investigated the mechanical and triological properties of PEEk under different irradiation condtion. I think this manuscript has many serious flaws, it should be well revised before publish in this journal.

The detail comments are as follows:

1. Please define the abbrivations when they first appears in the manuscript. For example, Line 17, PEEK1-7. Line 20, PEEK4. Line 36, PEFC.

2. Line 20, please give a specific data of wear rate and clarify the experiemtnal condition.

3. Compared other polymers, PEEK has good anti-irradiation performance. Please state clearly in the introduction part why you use this irradiation method to modify PEEK? The introduction part need to be improved.

4. The authors used the abbreviations such as PEEK1-7, PEEK4, and PEEK5 in the abstract, which will make the reader confused. Please state clearly about the samples so that the readers can compare with each other easily.

5. The references in the introduction part is not enough. Please update the references, cite more newly references, and well summarize the works what have been done. 

6.Please retake the triology test of PEEK-4. I think this curve is not reliable.

7.Error bar should be added in figure 2 and figure 4.

8.The authors should explain why prisitne PEEK, PEEK-1, PEEK-5, PEEK-6 increased sharply during the tests.

9. The authors claimed that there is some crosslinking in the PEEK matrix. But they did not do any characterization of this speculation.

10. There are many grammar mistakes in the whole manuscript.

Author Response

Reviewer 2. 

  1. Please define the abbrivations when they first appears in the manuscript. For example, Line 17, PEEK1-7. Line 20, PEEK4. Line 36, PEFC.

Thanks for pointing this out. All abbreviations of samples were rewritten by the values of obtained fluence and number of runs in the Abstract and Introduction parts of the manuscript.  Lines 21, 22 and 30, 31.

  1. Line 20, please give a specific data of wear rate and clarify the experiemtnal condition.

The detailed experimental conditions were provided in Materials and Methods section. Wear rates of Pristine PEEK and PEEK 4 were 13.1 *  and 4.57*  respectively, these exact values of ware rates provided in the Lines 21-22 of the abstract.

  1. Compared other polymers, PEEK has good anti-irradiation performance. Please state clearly in the introduction part why you use this irradiation method to modify PEEK? The introduction part need to be improved.

The study of radiation-induced effects on the mechanical, thermochemical, and structural properties of polymers is essential because high-temperature thermoplastic polymers, such as PEEK, are often used in harsh radiation environments [1]. According to literature results, both cross-linking and chain scission processes were observed simultaneously after high-energy electrons with an energy of 350 keV and doses of 12-34 MGy [2]. 1 MeV electron irradiation with fluences of 5*  e/  and 3*  e/  leads to a sharp reduction of breaking strength and a slight reduction of yield strength [3]. However, no studies have examined the mechanical, thermochemical, and structural properties of PEEK polymers after exposure to a relatively low dose of electron radiation. In this regard, we are interested in understanding how these properties of PEEK polymer might change as a result of irradiation with high-energy electrons at a relatively low dose (i.e., on the scale of hundreds of kGy).

These sentences were added to the article’s “Introduction” section.

  1. Manas, M. Ovsik, A. Mizera, M. Manas, L. Hylova, M. Bednarik and M. Stanek. The Effect of Irradiation on Mechanical and Thermal Properties of Selected Types of Polymers. Polymers 2018, 10, 158; [doi:10.3390/polym10020158]
  2. Rival, T. Paulmier, E. Dantras. Influence of electronic irradiations on the chemical and structural properties of PEEK for space applications. Polymer Degradation and Stability 168 (2019) 10894; [doi.org/10.1016/j.polymdegradstab.2019.108943]
  3. Li, J. Yanga, G. Lv, S. Dong, F. Tian, Sh. Donga, X. Li. In-Situ saxs/waxd analysis on structural evolution in peek irradiated by 1 MeV electrons during tensile deformation. Polymer Degradation and Stability 181 (2020) 10935; [doi.org/10.1016/j.polymdegradstab.2020.109350]

  1. The authors used the abbreviations such as PEEK1-7, PEEK4, and PEEK5 in the abstract, which will make the reader confused. Please state clearly about the samples so that the readers can compare with each other easily.

All abbreviations in the abstract were replaced with the exact irradiation dose numbers. Lines 21, 22 and 30, 31.

  1. The references in the introduction part is not enough. Please update the references, cite more newly references, and well summarize the works what have been done. 

The Introduction section rewritten according to comments.

  1. Please retake the tribology test of PEEK-4. I think this curve is not reliable.

Thank you for your suggestion. We did a new assessment of the tribology test with the previous experimental set up, which is shown in Figure 1.

  1. Error bar should be added in figure 2 and figure 4.

The error bars were added to the Figures 2, 4.

  1. The authors should explain why pristine PEEK, PEEK-1, PEEK-5, PEEK-6 increased sharply during the tests.

As the electron beam irradiation leads to the both cross linking and chain scission processes, it is essential to study the dependence of sample’s structural changes on irradiation parameters, such as total dose, number of runs and irradiation speed. The increase of the microhardness for those samples can be explained by the formed crystalline regions during irradiation. These explanations were reflected on lines 391-392.

  1. The authors claimed that there is some crosslinking in the PEEK matrix. But they did not do any characterization of this speculation.

The crosslinking behavior of PEEK samples were assessed by the Differential Scanning Calorimetry (DSC) results, where the high temperature shift of melting temperature for PEEK (Figure 7c in the manuscript) are brought about by formation of crosslinking [1].

  1. Sasuga, T.; Kudoh, H. Ion irradiation effects on thermal and mechanical properties of poly(ether–ether–ketone) (PEEK). Polymer 2000, 41, 185–194.
  2. There are many grammar mistakes in the whole manuscript.

A new version of the manuscript is well summarized and checked carefully for grammar mistakes.

Round 2

Reviewer 1 Report

The authors have responded adequately to the comments and changed the manuscript accordingly.

Reviewer 2 Report

The manuscript "Effects of electron beam irradiation on mechanical and tribological properties of PEEK" have improved a lot after the revision. But the authors still need to address the following comments.

1. Figure 1, the curves of PEEK-2 and PEEK-7 are quite different from other samples. These two curves are much smoother than others. I doubt the data reliability. Please give more detail explanation. 

2. The data in Table 2 is not right. You can only calculate the average friction coefficient after the sample enters into the steady state, which means the friction coefficient changes a little.

3. The Tg position in Figure 7b is not right.

4. The language should be carefully revised. There are too many mistakes.

Round 3

Reviewer 2 Report

I recommend it to publish in this journal.